# Secondary Minerals from Minothem Environments in Fragnè Mine (Turin, Italy): Preliminary Results

**Yuri Galliano** [1], **Cristina Carbone** [1,*], **Valentina Balestra** [2], **Donato Belmonte** [1] and **Jo De Waele** [3]

1. DISTAV, Università degli Studi di Genova, C.so Europa 26, 16132 Genova, Italy; yury.galliano@gmail.com (Y.G.); donato.belmonte@unige.it (D.B.)
2. DIATI, Politecnico di Torino, C.so Duca degli Abruzzi 24, 10129 Torino, Italy; valentina.balestra@hotmail.com
3. BiGea, Università di Bologna, Via Zamboni 67, 40126 Bologna, Italy; jo.dewaele@unibo.it
* Correspondence: cristina.carbone@unige.it

**Abstract:** The Fragnè mine, located in the Lanzo valley in the municipality of Chialamberto (Turin, Piedmont Region), represented the most important regional site for Fe–Cu sulfide exploitation over a period of more than eighty years (1884–1965). The entire mining area is part of a structural complex in the Lower Piedmont Unit of the Western Alps, characterized by the presence of amphibolite, metabasite ("prasinite"), and metagabbroic rocks. In particular, the pyrite ore deposit occurs as massive mineralizations within interlayered metabasites and amphibolites. In this work, we describe secondary minerals and morphologies of minothems from the Fragnè mine that are found only in abandoned underground works, such as soda straws, normal and jelly stalactites and stalagmites, jellystones, columns, crusts, blisters, war-clubs, and hair, characterized by different mineralogical associations. All minothems were characterized by minerals formed during acid mine drainage (AMD) processes. Blisters are composed only of schwertmannite, war-clubs by schwertmannite, and goethite with low crystallinity and hair by epsomite and hexahydrite minerals. Jelly stalactites and stalagmites are characterized by schwertmannite often in association with bacteria, while solid stalactites and stalagmites are characterized by jarosite and goethite. The results indicate that the mineralogical transformation from schwertmannite to goethite observed in some minothems is probably due to aging processes of schwertmannite or local pH variations due to bacterial activity. On the basis of these results, we hypothesize that all the jelly samples, in association with strong bacterial activity, are slowly transformed into more solid goethite, and are thus precursors of goethite stalactites.

**Keywords:** acid mine drainage; oxides-hydroxides; schwertmannite; goethite

## 1. Introduction

Secondary minerals forming speleothems, usually referred to as "cave minerals", are the result of complex interactions between bedrock, circulating water, and sediments of various sources [1]. A "speleothem" is a secondary mineral deposit formed in a natural cave [2] by a chemical reaction from a primary mineral assemblage in bedrock or detritus because of a unique set of conditions therein. An identification name is awarded to each particular type of speleothem and the subdivision into types is mainly based on morphological and genetic characteristics [1]. Usually, different morphologies correspond to different genesis; however, there are cases where two different genetic mechanisms lead to two indistinguishable morphologies. In this case, the two speleothems are attributed to the same type, describing the different mechanisms of origin. On the other hand, it is impossible to originate different shapes by the same formation process. A speleothem that resembles a certain type, but has a different genesis, is assigned to a subtype [1]. Most speleothems typically form from the precipitation of $CaCO_3$ (either calcite or aragonite) in caves developed in carbonate bedrock, but they can also be composed of other carbonates, oxides, hydroxides, sulfides, sulfates, phosphates, and silicates [1,3,4].

Hill and Forti (1997) [1] recognized that there are speleothems also in artificial caves. Carbone et al. (2016) [4] coined the name "minothem" including secondary mineral concretions forming in artificial underground voids, such as a mine or any other kind of man-made tunnel. Minothems are the counterpart of speleothems in natural caves, and generally show the same morphologies. However, the petrographical and geological differences of the host rock can cause significant differences in mineralogy, color, and shape of the minothems with respect to speleothems [4]

In this work, we characterize and describe secondary minerals and related minothems forming in Fragnè mine, Chialamberto (TO), Piedmont, Italy. The site is abandoned and is characterized by active and intense acid mine drainage (AMD) processes triggered by the supergene alteration of sulfide-rich mineralizations. Acid sulfate waters (ASW) percolating inside the galleries drip through the mine roof and form numerous decorative dripstone features that coat the walls, ceilings, and floors of the mine, and grow out of muck piles creating minothems. AMD minerals that commonly precipitate from these waters are Fe-oxy-hydroxides (such as, ferrihydrite, schwertmannite, jarosite, and goethite) and sulfates with variable chemical composition, depending on the associated host rocks and gangue minerals [4–7].

## 2. Materials and Methods

The Fragnè mine, Chialamberto (TO), Piedmont, Italy, is located in the Big Lanzo Valley, in the southernmost part of Monte Bellavarda, Graian Alps (Figure 1). The mine develops underground for about 5 km and has tunnels located on 11 levels, although many galleries are collapsed. It is located in the structural complex referred to as "Lower Piedmont Zone", which is made up of Mesozoic ophiolitic units of oceanic origin formed as a result of metamorphism from the ancient bottom of the Ligurian–Piedmont Basin [8]. In the study area, there are outcrops of the following lithotypes: serpentinites, metabasites (metagabbros and prasinites), and amphibolites [8]). The ore deposit consists of stratiform sulfide mineralizations characterized by the presence of twisted and folded lenticular bodies of massive pyrite and Cu-rich pyrite inside chlorite schists. The Cu-rich pyrite occurs in association with minor amounts of chalcopyrite, sphalerite, bornite, pyrrhotite, and galena.

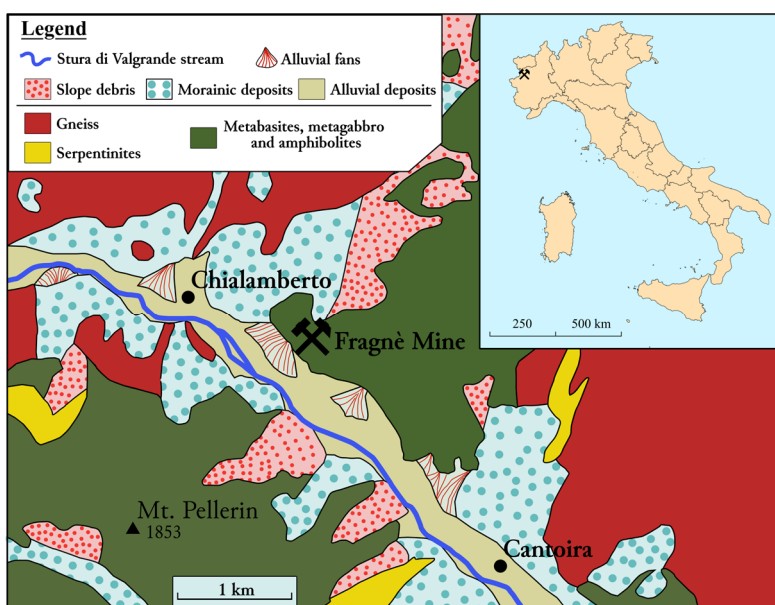

**Figure 1.** Geographical and geological sketch map of the Fragnè mine (Lanzo Valley, Western Alps). Modified after Foglio 41 Gran Paradiso (http://sgi.isprambiente.it (accessed on 22 June 2022)).

The sampling was performed together with speleologists from Liguria and Piedmont. All samples were taken from two lower levels, not clearly attributable to the old mining plans, due to the various collapsed galleries. These two levels are currently the only accessible ones. The first section of the entrance is flooded and characterized by AMD, with variable water depths depending on the seasons. The most extensive level is called Santa Barbara (899 m a.s.l.); the higher one is called Sobrero level (916 m a.s.l.).

Samples are classified with an identification code (level and progressive number), photos, and a brief description with macro- and microscopically details. All samples rangin in color with various hues of white, yellow, red, brown, green, and blue are representative of all types of minothems that occur inside the mine (Figure 2).

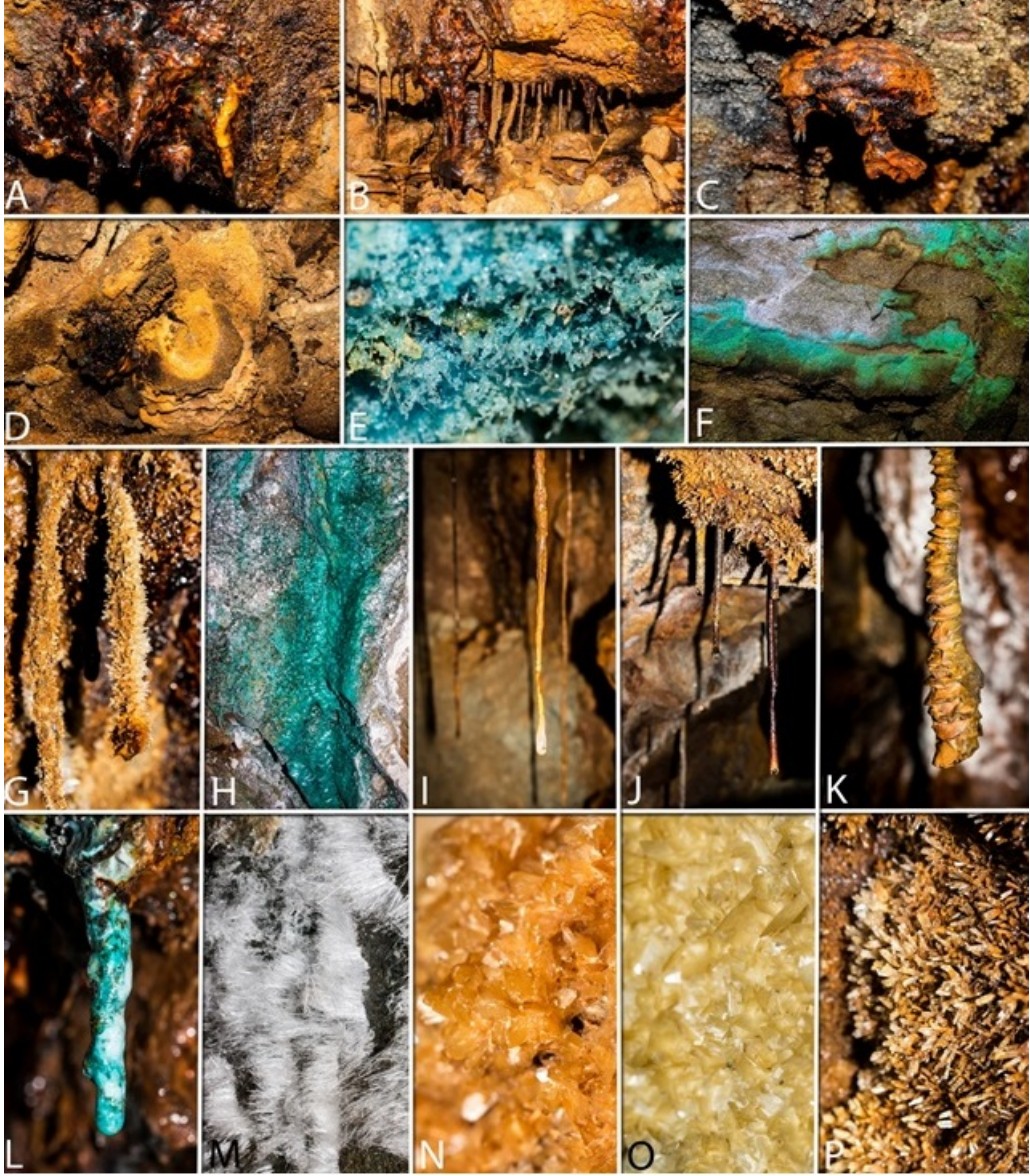

**Figure 2.** Minothems of Fragnè mine. (**A**) stalactite, Sobrero level; (**B**) column, Sobrero level; (**C**) blister, Sobrero level; (**D**) pancake stalagmite, S. Barbara level; (**E**) crust, S. Barbara level; (**F**) crust, S. Barbara level; (**G**) stalactite, Sobrero level; (**H**) jellystone, Sobrero level; (**I**) jelly stalactite, Sobrero level; (**J**) soda straw, S. Barbara level; (**K**) war club, Sobrero level; (**L**) jelly stalactite, S. Barbara level; (**M**) hair, S. Barbara level; (**N**) crystalline crust, S. Barbara level; (**O**) crystalline crust, S. Barbara level; (**P**) crust of needles, Sobrero level.

The terminology of minothems is according to Hill & Forti (1997) [1] and Carbone et al., (2016) [4]. Stalactites are common and present in different colors (Figure 2A,L) and are often covered with crystals (Figure 2G); stalagmites, not shown in Figure 2, are only dark brown to yellowish in color, and are generally of limited height (up to 5 cm). War club stalactites (Figure 2K) are about 3 cm in diameter and microgours cover their entire length. Varicolored jellystone covers roofs and walls (Figure 2H) and jelly stalactites (Figure 2I,J) show an upper (inner) harder part and a soft and gelatinous lower part. These samples were prepared for mineralogical analysis in order to observe a cross section in the growth direction, both in the upper and lower part. Blisters are found attached to coatings crusts or cave walls. Crusts cover walls, ceilings, and floor sediments and are often characterized by millimetric crystals with vitreous luster (Figure 2E,N,O,P). Moreover, a rather uncommon fibrous minothem called "hair" composed of crystal fiber aggregates occurs hanging from the walls of the mine (Figure 2M). These peculiar hair fibers reach 6 cm in length, with longer fibers breaking under their own weight. All samples were dried at room temperature for a few days. Mineralogical analyses were performed using X ray powder diffraction (XRPD) and scanning electron microscope with EDS analyses (SEM-EDS) at the DISTAV laboratories. XRPD was the most important method used for mineral identification; samples were ground in agate mortar and data measurements were carried out using a Philips PW 3710 diffractometer under the following conditions: start and end position (°2θ): 3–70; anode: Co; 2 Step Size (°Th): 0.0200 Scan Step Time (s): 1.0000, Scan Type: Continuous; Divergence Slit Type: Fixed Divergence Slit Size (°): 1.0000; data interpretation was carried out with the X'pert High Score software. SEM-EDS was useful to highlight morphological aspects and to perform analysis on the composition of what is observed by XRPD, i.e., precise micro-volumes of the sample. All samples were analyzed with a SEM Vega3—TESCAN type LMU, equipped with EDS detector APOLLO XSDD of EDAX with a DPP3 type analyzer at 15 kV accelerating voltage, 2–15 nA beam current, and 10–25 μm beam diameter associated with the TEAM EDS (Texture and Elemental Analytical Microscopy) software for the acquisition and processing of all data deriving from the analysis.

## 3. Results

The secondary minerals identified by XRPD are reported in Table 1. All samples are characterized by mineral species that typically form in AMD environments: Fe-oxy-hydroxides (mainly schwertmannite and goethite), but also sulfates such as gypsum, epsomite, hexahydrite, melanterite, jarosite, and ktenasite. The presence of chlorite, albite, quartz, and amphibole is attributable to the surrounding rocks, while pyrite is related to ore mineralizations. XRPD analysis showed very noisy diffraction patterns with broad peaks due to the poor crystallinity of the samples under study. Two types of poorly crystalline minerals were detected: allophane and schwertmannite. Allophane showed a diffraction pattern with two prominent bands at 3.35Å and 2.3Å, whereas schwertmannite showed its characteristic six bands. Moreover, goethite evidenced a very weak band, testifying a poorly crystalline state (Figure 3).

The XRPD results showed that schwertmannite and goethite were the main minerals occurring in gelatinous (jelly) and hard minothems. In order to understand the minerogenetic processes involved in the transformation between these two phases, cross-sections of jelly stalactites and hard stalactites were subjected to SEM investigations. All jelly stalactites and hard stalactites were characterized by concentric and rhythmic layers that develop around a large central feeding tube (Figure 4A). The inner zone is characterized by pin-cushion morphologies (Figure 4B), globular masses surrounded by radial fiber aggregates typical of schwertmannite and confirmed by EDS analysis. Images at high magnifications allowed identification of abundant bacterial structures (Figure 4C,D). The outer part was characterized by a compact zone with goethite grown in layers and with no signs of bacterial structures.

Table 1. Results of XRPD analysis on Fragnè mine samples.

| Sample Id Code | Level | Minothem | Minerals | PDF | Figure |
|---|---|---|---|---|---|
| Liv1_01 | Sobrero | Stalactite | Goethite, jarosite | 003-0249<br>010-0443 | 2A |
| Liv1_02 | Sobrero | Jellystone | Not detected | | 2H |
| Liv1_03 | Sobrero | Soda straw | Schwertmannite, poorly crystalline goethite | 047-1775<br>003-0249 | 2G |
| Liv1_04 | Sobrero | Column | Schwertmannite, poorly crystalline goethite | 047-1775<br>003-0249 | 2B |
| Liv1_05 | Sobrero | Warclub | Schwertmannite, poorly crystalline goethite | 047-1775<br>003-0249 | 2K |
| Liv1_06 | Sobrero | Jelly stalactite | Schwertmannite, poorly crystalline goethite | 047-1775<br>003-0249 | 2I |
| Liv1_07 | Sobrero | Blister | Schwertmannite | 047-1775 | 2C |
| Liv1_07p | Sobrero | Crystals on blister | Gypsum | 008-0467 | |
| Liv1_08 | Sobrero | Crust | Gypsum | 008-0467 | 2P |
| Liv3_01 | Santa Barbara | Jelly stalactite | Allophane | 002-0039 | 2L |
| Liv3_02 | Santa Barbara | Soda straw | Schwertmannite | 047-1775 | 2J |
| Liv3_04 | Santa Barbara | Hair | Epsomite, hexahydrite | 008-0467 001-0354 | 2M |
| Liv3_05b | Santa Barbara | Crust | Melanterite | 001-0255 | 2E |
| Liv3_07 | Santa Barbara | Pancake stalagmite | Poorly crystalline goethite | 003-0249 | 2D |
| Liv3_08a | Santa Barbara | Crust | Chlorite, amphibole, quartz, jarosite | 002-0028 01-073-1135<br>001-0649<br>010-0443 | |
| Liv3_08a | Santa Barbara | Crust | Albite, jarosite | 002-0515<br>010-0443 | |
| Liv3_08b | Santa Barbara | Crust | Gypsum | 008-0467 | 2N |
| Liv3_09 | Santa Barbara | Crust | Gypsum | 008-0467 | 2O |
| Liv3_10a | Santa Barbara | Crust | Gypsum, amphibole, pyrite, quartz, ktenasite | 008-0467<br>01-073-1135<br>003-0822<br>001-0649<br>029-0591 | 2F |
| Liv3_10b | Santa Barbara | Crust | Gypsum, amphibole, albite, jarosite | 008-0467<br>01-073-1135<br>002-0515<br>010-0443 | |

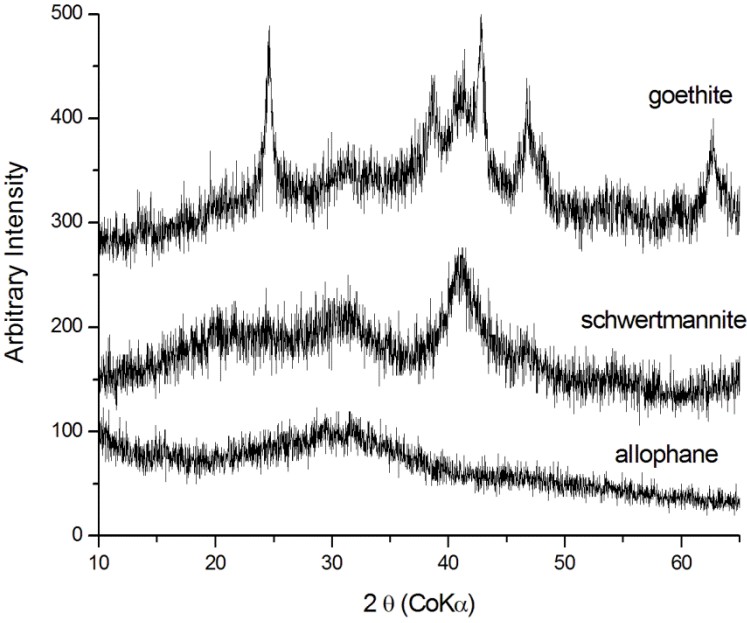

Figure 3. X-ray diffraction patterns of allophane, schwertmannite, and poorly crystalline goethite.

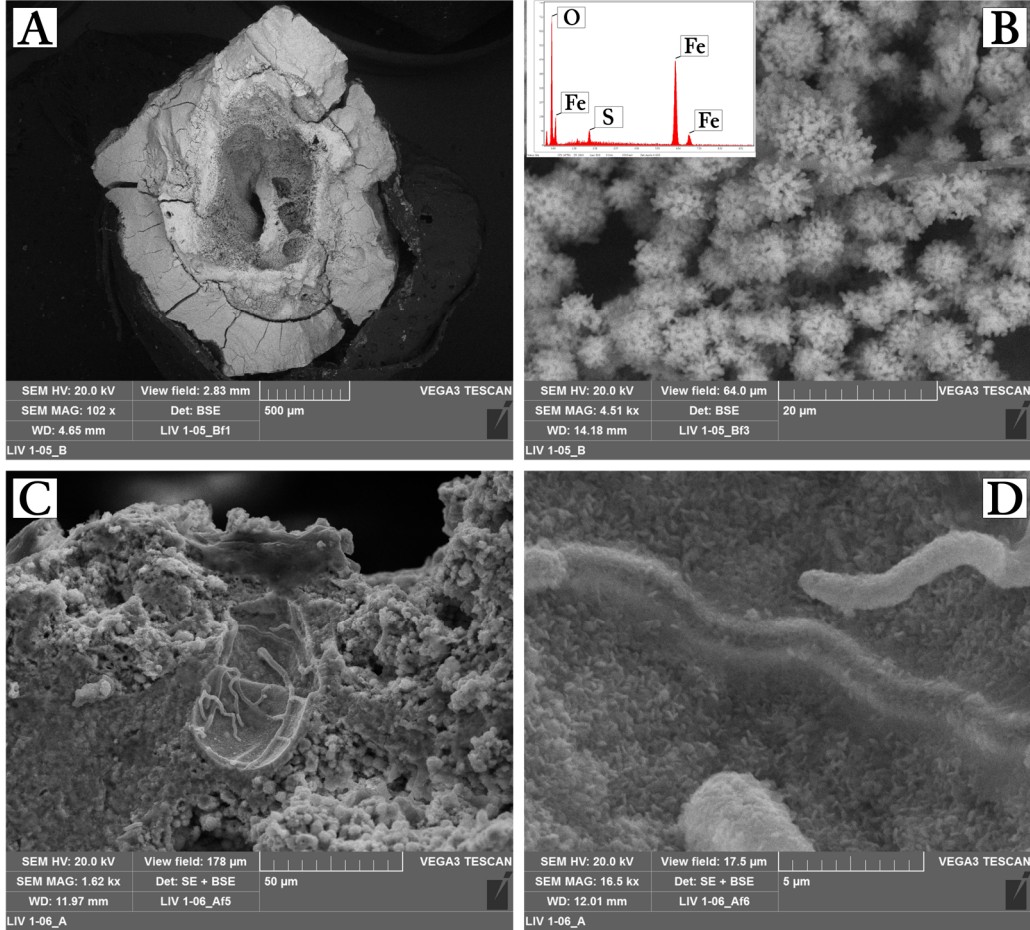

**Figure 4.** SEM image of a representative jelly stalactite. (**A**) General morphology showing the internal feeding tube. (**B,C**) Images of the internal younger jelly part of the stalactite, with schwertmannite aggregates (**B**) and microbial structures (**C,D**).

## 4. Discussion

The presence of Fe-rich phases (i.e., goethite, schwertmannite, and jarosite) in all minothems shows that the acid mine drainage (AMD) is still active at Fragnè Mine. The presence of gypsum, hexahydrate, and ktenasite further confirms the mobilization of chemical elements caused by acid drainage. In fact, the Mg- and Ca-rich-sulfates derive from the leaching of the minerals of the surrounding rocks, whereas Cu, Co, and Zn come from the mineralized masses. The evolution of schwertmannite versus goethite can be clearly observed in Figure 5, in which the XRD pattern corresponding to successive precipitate samples from the jelly (lowest pattern) to hard stalactites (highest pattern) are reported. In the initial stage, only schwertmannite was present, then with the evolution of transformation processes, goethite peaks tend to appear, becoming more pronounced in the final stage. Goethite was always in a poorly crystalline state. The variation of mineralogical composition of Fe-rich minothems could be due to the aging of schwertmannite, which, being a metastable phase, tends to transform into goethite [9–11].

Under acidic conditions, schwertmannite completely converts to a sulfate-bearing goethite within 100 days [12], while the rate of transformation is doubled when transferred to an environment under near neutral, oxygen-rich conditions [10,13]. This process is characterized by $OH^-$ consumption and by the release of $SO_4^{2-}$; the speed of release of these components depends on temperature and pH (being faster at high pH and/or high temperatures) [10] and on bacterial activity [14].

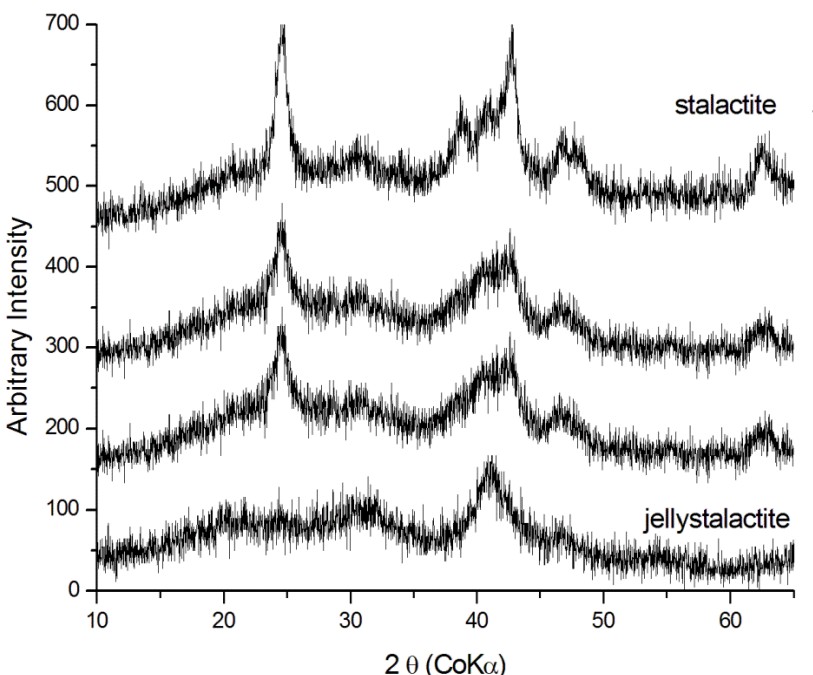

**Figure 5.** X-ray diffraction patterns for samples taken at different stages of evolution of representative samples of jelly stalactites (schwertmannite) and poorly crystalline goethite. The black arrow indicates schwertmannite into goethite transition.

The latter factor appears to be very relevant to explain the abundant presence of bacterial filaments. According to [14,15] the transformation of schwertmannite occurs in two stages through biochemical, chemical, and physical processes. The first phase is characterized by the reductive dissolution of schwertmannite mediated by bacteria that reduce iron or through sulfides produced from sulfate-reducing bacteria. The second phase is instead characterized by the formation of goethite through the transformation of the residual phase $Fe^-(OOH)$ or through the catalysis of $Fe^{2+}$. The phase variation inside the minothems is further confirmed by the XRPD results, from which it is possible to infer that the younger (inner) part of the jelly stalactite is characterized by schwertmannite turning into goethite, while the older (outer) part is almost totally made up of goethite. This process is probably speeded up by the high bacterial activity, but it is not possible to exclude that goethite derives from pH variations [16,17]. Jelly stalactites similar to those analyzed in this work and entirely composed of schwertmannite were found in some sulfide mines of Poland [18]. These Polish jelly stalactites show no evidence of goethite, which might indicate that they are younger, or that their transformation is slower due to the lower temperatures. On the contrary, schwertmannite was found as interlayer crusts in phyllites and in a stalactite sample taken in Flaschar's Mine, Czech Republic [19]. Jelly stalactites with the composition of our minothems are very difficult to find as speleothems in natural caves: only a particular case was reported by [17] in the gallery of Queimada, a basaltic tube located in the center of the island of Terceira (Azores). Iron-oxide bearing speleothems were found in Lechuguilla Cave in New Mexico, but were entirely composed of well-crystallized radiating goethite, grown around a central bacterial tube [19]. These authors retain that goethite evolved from a poorly crystalline precursor phase over long periods of time. Goethite stalactites up to half a meter long have been described from Vărai Cave in the Purcăreț-Boiu Mare Plateau in northern Romania and derive from oxidation of pyrite [20]. It is worth emphasizing that minothem Fe-oxy-hydroxide stalactites grow from the inside; i.e., the youngest layers surround the feeding channel, and these growing schwertmannite layers push the older (outer), more mature Fe-oxy-hydroxide layers outward. Also, jelly stalactite tips are composed of younger schwertmannite. The external (older) layers, when the transformation is complete, are composed of goethite. This is in contrast with carbonate

stalactites (speleothems), where the older layers are the ones surrounding the feeding channel, and water films degassing $CO_2$ form younger carbonate layers on their outside or at their tips. Because of these different mechanisms of formation, probably the use of "stalactite" (or jelly stalactite) in mines should be avoided, or substitutes by "stalactite-like" minothems, in which "stalactite" only refers to the shape of the minothem. Further studies will be carried out in order to investigate these minerogenetic aspects.

## 5. Conclusions

Minothems at Fragnè mine reveal different shapes including stalactite-like shapes, stalagmites, columns, crusts, soda straws, warclubs, blisters, and hair forms. We studied forms not yet described in mining environments and that were found for the first time in the Fragnè mine, including jelly stalactites and jelly stalagmites. Jelly stalactites are particular stalactites that have a gelatinous consistency in the most recently deposited part (along the inner feeding tube and at their tips). Some are made up of a soft inner (central) part of schwertmannite with bacterial filaments and outer layers of goethite; others instead have a less-clear division between the hard and the soft part and are composed of poorly crystallized allophane, iron, and manganese oxides and hydroxides. Jelly stalagmites are particular stalagmites that have a gelatinous consistency in the more recently deposited part and are constituted of goethite in their inner (older) parts. The association of goethite and schwertmannite found in some samples is probably due to the aging of the schwertmannite, because this metastable phase tends to transform into goethite over time. Fe-oxy-hydroxide stalactite-like minothems thus grow from their inside, with the youngest layers surrounding their central feeding channel and their tips, pushing out the outer (older) layers of goethite.

**Author Contributions:** Conceptualization, Y.G., C.C. and J.D.W.; methodology, V.B., C.C.; software, C.C.; validation, Y.G., D.B., V.B.; investigation, V.B., C.C.; resources, D.B., J.D.W.; data curation, Y.G., C.C.; writing—original draft preparation, Y.G., V.B., C.C.; writing—review and editing, D.B., J.D.W.; supervision, C.C.; funding acquisition, D.B., C.C. All authors have read and agreed to the published version of the manuscript.

**Funding:** This research received no external funding.

**Data Availability Statement:** Not applicable.

**Acknowledgments:** The authors are very grateful to Marco Barchi and Marco Sacchi of Gruppo Speleologico EXPLORA of Lanzo C.A.I. for accompanying some of us in the Fragnè mine, to Enrico Lana of Biologia Sotterranea Piemonte—Gruppo di Ricerca for helping us bring samples.

**Conflicts of Interest:** The authors declare no conflict of interest.

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
