# Peer review of "Secondary Minerals from Minothem Environments in Fragnè Mine (Turin, Italy): Preliminary Results"

_minerals, doi:10.3390/min12080966_

Round 1
Reviewer 1 Report
General Comments:
This study was caried out by authors who coined the term “minithem”. There are just a handful of articles about minothems, so this study is a welcome addition to the topic. Unfortunately, the manuscript remained at the level of presenting the mineral composing the minothems, based on x-ray powder diffraction data and EDS, but for the later one, data is not provided.
Any further “ideas” about the minerals formation remain just suppositions, because no data are provided to constrain the formation conditions. The mentioned transformation process due to ageing or pH is just a possibility, but no data is provided at least to boost the likelihood of happening theory.
In the “Materials and Methods” section, starting from lines 96-101 contains sample descriptions that need to go to section “Results”.
There is need for details on sample preparation for XRPD and SEM due to water content and hardness.
The authors need to provide details on PW 3710 setup (2theta interval, step, counting time/step, X-ray source kV/mA and optics (filters, monochromators etc.)
Why authors of this study not considered ammonium oxalate extraction for elimination of amorphous fazes?
Why authors did not provide environmental data: temperature, humidity, drip water pH, EC, Eh, drip water chemistry, CO2 level in air?
For each mineral, the matched powder diffraction file number should be provided (ex. PDF 47-1775)
The authors state (lines 175-179) that the outer part of the jelly stalactite is composed of schwertmannite and the inner (older part) totally made up from goethite but on lines 137-144, the author presents the contrary. This contradiction needs explanation.
Additional comments:
In the introduction section, the authors are encouraged to provide some reference for secondary minerals related to AMD and the environmental conditions.
Line 19: Change jelllystones with jellystones.
Lines 24-26: The mentioned transformation process due to ageing or pH is just a possibility, but no data is provided.
Line 45: The number 3 should be formatted with subscript.
Lines 65-66: There is no need for web link, pleas remove it. The webpages are changing way to fast and are not a reliable source of information.
Line 77: Change “) Modified by Caso et al., 2021.” to “). Modified after [5].” or “). Map based on [5] data.”.
Line 79: Replace “in” with “from”.
Lines 79-83: A sketch presenting a cross section with the mine adits.
Line 86: The colour description would be more appropriate using “various hues of white, yellow, red, brown, green and blue “. Replace color with colour.
Lines 111-112: Replace “PW 3710 with Philips X'pert High Score as software of data interpretation” with “measurements were carried out using Philips PW 3710 diffractometer and data interpretation with X'pert High Score software.”.
Lines 125-126: The correct expression is “diffraction patterns with broad peaks”.
Lines 127-129: allophane showed a diffraction pattern with two prominent bands at 3.35Å and 2.3Å whereas schwertmannite showed its classic 6 bands. Moreover, also goethite evidenced a very weak band testifying a nano-crystalline state
Line 130: Please delete the second “Table 1”.
In Table 1, the second row, the mineral column contains “Amorphous”, which is not a mineral!
Probably the column “Mineral” may be replaced by “Material” then the “Amorphous” is ok.
Line 132: On X axis the 2 theta is missing and the CoKa must be in (CoKa) form. On figure 3, the goethite (Gth) and schwertmannite (Swm) peaks need to be marked. The allophane is short-range ordered not amorphous.
The “amorphous” faze can hide small crystalline fazes. Using ammonium oxalate, the “amorphous” faze can be leached and increase the crystalline faze quantity, that can be detected with X-ray diffraction.
Line 154: Replace “XRD spectra” with “XRD pattern”.
Line 160: Delete “(“.
Line 162: On X axis the 2 theta is missing and the CoKa must be in (CoKa) form. On figure 5, the goethite (Gth) and schwertmannite (Swm) peaks need to be marked. What authors consider a solid argument for transformation and not just a compositional variation. What samples were used for this XRD patterns.
Line 170: Replace “Bao et al. (2018)” with [11].
Lines 175-179: The authors state that the outer part of the jelly stalactite is composed of schwertmannite and the inner (older part) totally made up from goethite but on lines 137-144, the author present the contrary. Which is true?
Line 193: The “Vārai Cave in the Purcāret-Boiu” correctly is “Vărai Cave in the Purcăreț-Boiu”
Line 240-254: Line indent missing.
Line 254: Correctly is “Tămaș”.
Author Response
In attached file our reply

Reviewer 2 Report
In this paper, the authors have presented the preliminary characterization of secondary minerals identified the abandoned Fragnè Mine (Turin, Italy). The paper presents a multi-method approach to characterize secondary minerals in minothem. Some issues should be addressed in the revised version.
The methodological approach provides a adequate viewpoint on the new discoveries. However, the complete description of the analytical protocol is missing. The experimental details should be extensively described in specific paragraph, so that other scientists can reproduce and repeat the experiments. Results and discussion of data are well detailed. The comprehensive discussion of data helps the reader to understand the impact of the present work and place these new discoveries in a broad contest. Overall the quality of the study is good. I suggest the publication of the paper with major revision.
Please consider the comments below to revise your manuscript:
- Line 28. What do you mean by “consistent” in this context?
- Line 49-54. Please change the sentence as follow. Hill & Forti (1997) defined the term minothem, which are the counterpart of speleothems in natural caves. Carbone et al. (2016) extended the definition of “minothem” including secondary mineral concretions forming in artificial underground voids, such as a mine or any other kind of man-made tunnel.
- Line 55. Delete for the first time
- Line 96. Change “present in” with “show”
- Line 97. “covered with crystal” is too generic. What kind of crystals? Have you identified them?
- Line 102. “for analysis” is not clear. What analysis did you prepare the samples for?
- Line 105. “crystal with vitreous luster” is too generic. What kind of crystals are they? Have you identified them?
- Line 109. What are the experimental conditions you used for performing the XRD analysis? What is the X-ray source (Cu, Co, or Mo)?
- Line 110. What are the experimental conditions you used for performing the SEM-EDS analysis?
- Line 125. Change “simple” with “noisy”
- Line 126. Add after “bands” the following “due to the poor crystallinity of the samples under study”
- Line 128. Change “classic” with “characteristic”
- Line 128. The peak is too broad to define it nano-crystalline. Please change with poorly crystalline state.
- Line 154-155. The sentence is unclear. Please rephrase. What do you mean by have been ordered from the jelly?
- Line 158- change “nano-” with “poor-”
- Line 159. Change “is probably” with “could be”
Author Response
in attached file our reply

Round 2
Reviewer 1 Report
1. We now inserted one EDS spectrum in fig 3.
I see no additions (EDS) to the fig 3.
2. Due to the collapsed tunnels and some very narrow tunnel connection environments, the 3d survey took longer than expected. The 3d data processing will take further time. Therefore, at the moment it is not yet possible to use the 3d survey of the mine.
The main point was to have additional information on the sampled site. How far from entrance are this samples taken.
3. At Line 115: “samples were pulverized with an agate mortar and data measurements”
Suggest “ground in agate mortar”
4. In line 117
replace 2Th replace with 2θ
5. Lines 173 “and stalactites (poorly goethite). “
Replace with “poorly crystalline goethite”
6. Reference [20] Correctly is “Tămaș” (is missing the last character)
Author Response
I follow your suggestion anc change. I insert EDs spectra in fig 3

Reviewer 2 Report
The authors took my proposed changes into account when producing the revised version and as for these reason, I suggest the publication of the paper in the present form.
Author Response
thank you for your suggestions and comments.
